# Kinetic Study of the Atmospheric Oxidation of a Series of Epoxy Compounds by OH Radicals

Carmen Maria Tovar,*[1] Ian Barnes[1†], Iustinian Gabriel Bejan*[2] and Peter Wiesen[1]

[1] Institute for Atmospheric and Environmental Research, University of Wuppertal, Wuppertal, 42097, Germany
[2] "Alexandru Ioan Cuza" University of Iasi, Faculty of Chemistry and Integrated Center of Environmental Science Studies in the North-Eastern Region – CERNESIM, Iasi, 700506, Romania
[†] deceased, on January 1, 2018

*Correspondence to*: Carmen Maria Tovar (carmen.tovar_ramos@uni-wuppertal.de) and Iustinian Gabriel Bejan (iustinian.bejan@uaic.ro)

**Abstract.** The kinetics of the gas-phase reactions of hydroxyl radicals with cyclohexene oxide (CHO), 1,2-epoxyhexane (EHX), 1,2-epoxybutane (12EB), *trans*-2,3-epoxybutane (*t*EB) and *cis*-2,3-epoxybutane (*c*EB) have been investigated using the relative rate technique. The experiments have been performed at (298±3) K and (760 ± 10) Torr total pressure of synthetic air using different reference compounds in a 1080 *l* Quartz Reactor (QUAREC) and a 480 *l* Duran glass chamber. The following room temperature rate coefficients (cm$^3$ molecule$^{-1}$ s$^{-1}$) were obtained: $k_1$ (OH + CHO) = (5.93±1.78) × 10$^{-12}$, $k_2$ (OH + EHX) = (5.77±1.29) × 10$^{-12}$, $k_3$ (OH +12EB) = (1.98±0.39) × 10$^{-12}$, $k_4$ (OH + *c*EB) = (1.50±0.26) × 10$^{-12}$, $k_5$ (OH + *t*EB) = (1.81±0.42) × 10$^{-12}$. With the exception of previous studies for 1,2-epoxybutane and cyclohexene oxide, this is to the best of our knowledge the first kinetic study of the reaction of these compounds with OH radicals. Atmospheric lifetimes, reactivity trends and atmospheric implications are discussed considering the epoxy compound rate coefficients obtained in the present study. In addition to a direct comparison with the literature data where possible, the results from the present study are compared with values estimated from the Structure Activity Relationship method.

## 1 Introduction

Oxygenated volatile organic compounds (OVOCs) play an important role in atmospheric chemistry and have an impact on climate and human health (Calvert et al., 2011). From those OVOCs emitted from either biogenic or anthropogenic sources, cyclic ethers, with the exception of furans (Villanueva et al., 2009; Li et al., 2018) have received very little attention. Epoxides, as simplest cyclic ethers, are an important and valuable class of raw materials and intermediates for chemical industry. They can polymerize for the production of homo- and copolymers as polyethers, polyols and polycarbonates (Hereijgers et al., 2012).

Epoxides are considered a key element in "click-chemistry" (Kolb et al., 2001; Fokin and Wu, 2006). They are also relevant in the field of pharmaceutical applications because of their potential as protease inhibitors against several diseases like cancer,

stroke, and parasitic or viral diseases (Powers et al., 2002; Otto and Schirmeister, 1997; Schirmeister and Klockow, 2003). A relatively new utilization of epoxides, which could have huge atmospheric implications and an impact on climate change is their potential use for carbon capture and storage, i.e. the chemical fixation of $CO_2$ in the form of cyclic carbonates in the presence of various catalysts (Zou and Hu, 2017; Zhang et al., 2020; Guo et al., 2021; Andrea and Kerton, 2021; Appaturi et al., 2021).


Epoxides are known to be formed from the reaction of $O(^3P)$ with isoprene at nearly 80% yield (Atkinson et al., 1994b; Paulson et al., 1992). In the atmospheric ozonolysis of isoprene epoxide yields of about 2-3% have been observed (Atkinson et al., 1994a). A few percent of epoxide formation have been reported from α-pinene ozonolysis as well as from the reaction of 1,2-dimethyl-1-cyclohexane with ozone (Alvarado et al., 1998). Atkinson et al. (1994a) concluded that epoxides formation is a process occurring in the ozonolysis of most terpenes and cycloalkenes. Low yields of isoprene derived epoxides have been observed also from the gas phase nitration of isoprene (Skov et al., 1994).


Furthermore, in indoor pollution studies the formation of epoxides from the heterogeneous oxidation of VOCs with gas phase ozone has been reported (Zhou et al., 2017). Other compounds could also form epoxides during their gas phase chemical degradation by OH radicals and ozone. More recent theoretical and mechanistic studies report epoxide formation during the gas phase ozonolysis of methylbutenol and sabinene via the Criegee degradation mechanism (Calvert et al., 2000; Almatarneh et al., 2019a, b).


Epoxides are responsible for up to 10% of the open ring products from the degradation of aromatic hydrocarbons in the presence of OH radicals according to the MCM model (Master Chemical Mechanism) (Jenkin et al., 2015). In the past, benzene oxide has been reported as an epoxide type product in the reaction of benzene with OH radicals (Klotz et al., 1997, 2000). Besides, a theoretical study has reported epoxide formation in the OH radical initiated oxidation of dimethylphenol isomers (Sandhiya et al., 2013).


Secondary organic aerosols (SOA) constitute a substantial portion of the total ambient aerosol particles, which are mainly originating from biomass burning and atmospheric reactions of volatile organic compounds (VOCs) (Kanakidou et al., 2005). Epoxides can polymerize easily, leading to the growth of SOA (Gao et al., 2004). A recent study shows that the yield of epoxides from the reaction of aromatic compounds with OH radicals is probably dependent on the abundance of NO, $HO_2$ and $RO_2$ in the reaction system (Vereecken, 2018). This aspect represents an uncertainty in atmospheric models to reliably estimate the expected SOA formation from reactive epoxide uptake by aerosols (Paulot et al., 2009). The presence of epoxides in SOA are highly probable for conditions prevailing in the atmosphere leading to higher hygroscopicity of particles with potential cloud condensation nuclei activity by conversion to organosulfates through an acid catalysed mechanism (Fuzzi et al., 2015).



Isoprene derived epoxides are very important intermediates, which could explain at least in part the composition of ambient SOA both in urban areas (Lin et al., 2013) and remote regions (Jacobs et al., 2013; Stropoli et al., 2019; Paulot et al., 2009; Jacobs et al., 2013; Shrivastava et al., 2019).

The toxicity of ultrafine particles such as SOA is not only related to their atmospheric concentration but also to the nature and chemical properties of both the precursors and the formed SOA components (Jiang et al., 2019). In this sense, epoxides are of

great concern because the epoxy functional group can act as an electrophile in its interaction with DNA and nucleosides, producing carcinogenic and mutagenic damages (Ehrenberg and Hussain, 1981).

The kinetic and mechanistic database available on the gas-phase reactions of epoxy compounds with the atmospheric oxidants OH and $NO_3$ radicals, $O_3$, and chlorine atoms are very scarce. For the compounds investigated in the present study, there is only one previous relevant study towards chlorine atoms at 298K (Tovar et al., 2021) . Two studies have reported the rate

coefficients of the reaction of 1,2-epoxybutane towards OH radicals (Wallington et al., 1988a; El Othmani et al., 2021a) and more recently cyclohexene oxide with OH radicals have been measured as a function of temperature (El Othmani et al., 2021a). In the present study we have performed for the first time a kinetic study of the reaction of OH radicals with 1,2-epoxyhexane, *trans*-2,3-epoxybutane and *cis*-2,3-epoxybutane at 298 K using the relative rate method.

Since several of the rate coefficients for the reactions of epoxides with OH radicals have been measured in the present study

for the first time and cannot be compared with literature values accordingly, we have applied different approaches for the estimation of structure-activity relationship (SARs) for the compounds studied in this work.

## 2 Methods

The experiments were performed in two different environmental chambers, namely a 1080 *l* quartz-glass photoreactor and a 480 *l* Duran glass chamber. These two photoreactors are briefly described below.

### 2.1 480 *l* reactor

This reaction chamber consists of a cylindrical Duran glass vessel (3m length, 45 cm dia.) closed at both ends by Teflon coated aluminium end flanges. Integrated on the metal flanges are ports for the inlet of reactants into the chamber and for the collection of samples from the reaction mixtures for further analyses. Other accessories, like a mixing fan to ensure homogeneity of the reaction mixtures and a capacitance manometer, are also located on the flanges. Arranged concentrically around the outside of

the reactor are 32 super actinic fluorescent lamps (Philips TLA 40 W, $300 \leq \lambda \leq 460$ nm, $\lambda_{max} = 360$ nm. The vacuum (ca. $10^{-3}$ mbar) is maintained by means of a Leybold turbo-molecular pump, model RUVAC WZ 151 (500 m$^3$/h), backed by a Leybold double stage rotary fore pump, model D40B (200 m$^3$/h). A White-type mirror system mounted inside the reactor is set to an overall optical path length of 51.6 m. The analysis of the reactants was done, during the experiments by *in situ* FTIR (Fourier Transform Infrared) long path spectroscopy using a resolution of 1 cm$^{-1}$. The FTIR spectrometer (Nicolet Magna 520) and the

transfer mirror system are covered with a protective box and are permanently purged with dry air to remove water vapour. The spectrometer is directly controlled by the OMNIC software, which is provided by Nicolet.

### 2.2 1080 *l* reactor

A detailed description of the reactor can be found in the literature (Barnes et al., 1994). Briefly, the reactor consists of the two quartz glass tubes with an inner diameter of 47 cm and a wall thickness of 5 mm. The reactor has a total length of 6.2 m.

Silicone rubber rings are used for all the glass-metal connections as well as for metal-metal connections. The reactor is connected to a turbo molecular pump system by which an end vacuum of $10^{-3}$ mbar can be achieved. A total of three fans are used for homogeneous mixing of compounds within the reactor. Different types of inlets are mounted on the end flanges for the introduction of chemicals and for pressure measurements.

The beam from an FT-IR spectrometer is coupled via a transfer mirror system into the reactor through KBr windows located
in one of the end flanges. A White-type mirror system (base path length (5.91±0.01) m), mounted inside the reactor, is used for multiple-reflection of the infrared beam within the reactor before it reaches the detector. Reactants were monitored *in situ* in the reactor in the infrared using 82 traverses of the beam, which is equivalent to a total optical path length of (484.7±0.8) m. All spectra in this work were recorded with a spectral resolution of 1 $cm^{-1}$. The FT-IR spectrometer NICOLET NEXUS was used, which is equipped with a liquid nitrogen cooled (77 K) mercury-cadmium-tellurium (MCT) detector. A Globar was
used as IR light source. All mirrors are gold coated to achieve optimum reflectivity.

Two different types of lamps (32 each) are installed around the reactor. They are mounted alternatively around the reactor to ensure homogeneity of the light intensity within the reactor. The first type, 32 superactinic fluorescent lamps (Philips TL05 40W: 300– 460 nm, max. intensity at ca. 360 nm) and 32 low-pressure mercury vapour lamps (Philips TUV 40W: max. intensity at 254 nm) can be used to irradiate the reaction mixture.
Typically, 64 interferograms were co-added per spectrum over a period of approximately 1 minute and 15-20 such spectra were recorded per experiment.

## 2.3 Relative Rate Method

The relative rate method was used to determine the rate constant of the OH radical induced oxidation of the epoxy compounds. The photolysis of $CH_3ONO$ in the presence of NO was used to produce OH radicals:

$$CH_3ONO + h\nu \, (\lambda \sim 360 \text{ nm}) \rightarrow CH_3O + NO$$
$$CH_3O + O_2 \rightarrow CH_2O + HO_2$$
$$HO_2 + NO \rightarrow NO_2 + HO$$

The concentrations or mixing ratios were measured from the exact amount of the compound added to the chamber. The liquids and gases have been injected into the chamber via a septum using a gas-tight syringe in a flow of air with the reactor being under reduced pressure. The glass-steel inlet system has been heated up to 60ºC to ensure the addition of the entire amount of epoxides and reference compounds into the reactor. After addition and filling the chamber to 760±10 Torr pressure of synthetic air, up to 15 infrared spectra (15 min) have been recorded to prove the constant concentration of the compounds in the chamber.
These tests have been performed not only to check if there is a diffusion of the injected compounds from the inlet into the chamber but also to evaluate the potential wall deposition of the added compounds in order to correct the rate coefficients for additional loss processes. Any diffusion from the inlet would affect the linearity of the kinetic plots, which otherwise would

not have been obtained. The different epoxides and reference compounds employed in the present study did not show a wall loss during the time of the experiment.

In the presence of the OH radical the corresponding epoxide and reference compound are consumed by the following reactions:

$$OH + Epoxides \rightarrow \quad Products \quad (k_{Epoxide}) \ (1)$$
$$OH + Reference \rightarrow \quad Products \quad (k_{Reference}) \ (2)$$

Provided that the reference compound and the epoxide are lost only by reactions (1) and (2), then it can be shown that:

$$ln\left\{\frac{[Epoxide]_0}{[Epoxide]_t}\right\} = \frac{k_{Epoxide}}{k_{Reference}} ln\left\{\frac{[Reference]_0}{[Reference]_t}\right\} (3),$$

where, $[Epoxide]_0$, $[Reference]_0$, $[Epoxide]_t$ and $[Reference]_t$ are the concentrations of the corresponding epoxy compound and
reference compound at times $t=0$ and $t$, respectively and $k_{Epoxide}$ and $k_{Reference.}$ are the rate coefficients of reactions (1) and (2), respectively.

To test for a possible loss of the reactants through photolysis, mixtures of the reactants in air in the absence of methyl nitrite were irradiated for 30 min and photolysis were found to be negligible for both the epoxide and the reference compounds. Additionally, various tests were performed to assess possible loss of the reactants via deposition on the chamber walls, and no
significant wall loss of the epoxy compounds and references was observed leaving the compounds in the dark in the reactor.

A minimum of two experiments for each epoxide compound has been performed in this study and up to three references compounds have been used. Reference compounds have been chosen based on similar reactivity as epoxides and suitability for FTIR subtraction procedures. Conversion of epoxides and reference compounds through the reaction with OH radical has been achieved up to 50%. Initial mixing ratios used in the 1080 $l$ reactor for the epoxides and reference compounds have been
as follow (in ppmv with 1 ppmv = 2.46 x $10^{13}$ molecule cm$^{-3}$ at 298 K and 760 Torr of total pressure of synthetic air): epoxides between 3-6 ppmv and reference compounds between 1 and 4 ppmv as shown in

Table 1. Concentrations used for the reactions performed in 480 $l$ reactor have been up to 8 times higher. Around 8 ppmv CH$_3$ONO has been added in 1080 $l$ reactor and up to 10 times more in 480 $l$ reactor. All epoxy and reference compounds used in this study were obtained from Sigma Aldrich and used without further purification. The stated purities were as follows:
cyclohexene oxide, 98%; 1,2-epoxyhexane, 97%; 1,2-epoxybutane, 99%; *trans*-2,3-epoxybutane,97%; for *cis*-2,3-epoxybutane, 97%; propylene, 99%; butane, 99%; *iso*-propyl acetate, 99.6%; *sec*-butyl acetate, 99%; ethylene, 99.5% and 99.985% for synthetic air, respectively, which was from Messer.

**Table 1.** Initial mixing ratios used in the 1080 *l* reactor and 480 *l* reactor for the epoxide and reference compounds in ppmv (1 ppm = 2.46 x $10^{13}$ molecule cm$^{-3}$) at 298 K and 760 Torr of total pressure. The mixing ratios have been estimated from injections of the compounds into the reaction chambers and provide information about the amount of compounds used in this study.

| Compounds | Initial mixing ratios (ppmv) | |
| --- | --- | --- |
| | *Reactor* (**1080 *l***) | *Reactor* (**480 *l***) |
| *Epoxides* | | |
| cyclohexene oxide | 6 | 17 |
| 1,2-epoxyhexane | 6 | 28 |
| 1,2-epoxybutane | 6 | 23 |
| *cis*-2,3-epoxybutane | 3 | 28 |
| *trans*-2,3-epoxybutane | 3 | 23 |
| | | |
| *Reference compounds* | | |
| Propylene | 4 | 8 |
| Butane | 4 | - |
| *iso*-propyl acetate | 1 | 9 |
| *sec*-butyl acetate | 1 | 9 |
| Ethylene | 4 | 21 |
| *methyl nitrite (OH radical precursor)* | 8 | 84 |

## 3 Results and Discussion

The experimental data from the kinetic experiments are plotted according to equation (3). Figure 1 shows the results obtained for the rate coefficient determinations from the OH radical initiated oxidation of: cyclohexene oxide using ethylene, propylene and isobutene as reference compounds; 1,2-epoxyhexane using ethylene and propylene as reference compounds; 1,2-epoxybutane using ethylene and iso-propyl acetate as reference compounds; trans-2,3-epoxybutane using sec-butyl acetate, ethylene and propylene as reference compounds, and cis-2,3-epoxybutane using sec-butyl acetate, iso-propyl acetate and butane as reference compounds. The selection of the reference compounds employed in the kinetic study has been constrained by the experimental technique since there is required to identify the infrared absorption bands which do not overlap with other infrared absorptions features from reaction mixture. All plots showed very good linearity despite the difficulties from handling the epoxides and subtracting the IR spectra.

The rate coefficients $k_{Epoxide}$ given in Table 2 were put on an absolute basis using the following values for the reactions of the reference compounds (cm$^3$ molecule$^{-1}$ s$^{-1}$): k(OH + propylene): $(2.44\pm0.37) \times 10^{-11}$ (IUPAC Task Group on Atmospheric Chemical Kinetic Data Evaluation, 2021), k(OH + ethylene): $(8.52\pm0.13) \times 10^{-12}$ (Calvert et al., 2015), k(OH + isobutene): $(5.10\pm1.33) \times 10^{-11}$ (IUPAC Task Group on Atmospheric Chemical Kinetic Data Evaluation, 2021), k(OH + *iso*-propyl

acetate): (3.72±0.29) ×10$^{-12}$ (Wallington et al., 1988c), k(OH + *sec*-butyl acetate): (5.65±0.59) × 10$^{-12}$ (Wallington et al., 1988c), k (OH + butane): (2.38±0.24) × 10$^{-12}$ (McGillen et al., 2020).

The rate coefficients obtained by using two simulation chambers are in perfect agreement as shown in **Figure 2**, which is a comparison of the data for EHX using propylene as reference compounds. It is fair to mention that for the epoxides where the opportunity was had to carry out the kinetic studies in both reactors, the linearity was as shown for this case. Both for OH

radicals and with chlorine atoms. Therefore, considering they are the same family of compounds, it can be concluded that the results are reproducible in both reactors. Accordingly, the data have been plotted together for all the epoxides regardless the reactor where the experiment has been performed.

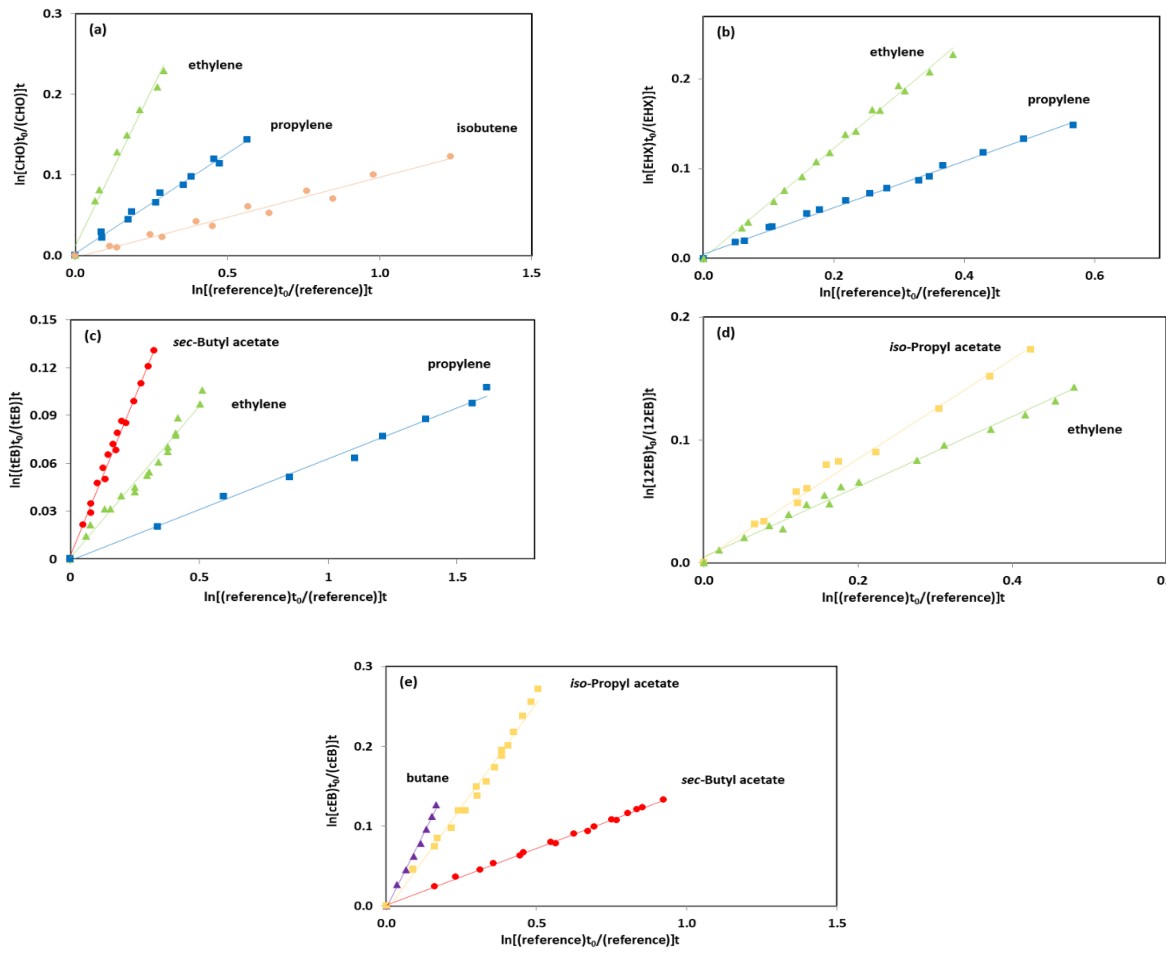

**Figure 1.** Kinetic data for the reaction of OH radicals with (a) CHO; (b) EHX; (c) *t*EB; (d) 12EB; (e) *c*EB using ethylene, butane, *sec*-butyl acetate, *iso*-propyl acetate and propylene as reference compounds.

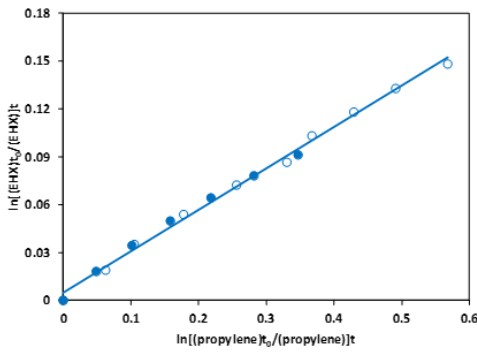

**Figure 2**. Kinetic data for the reaction of OH radicals with EHX using propylene as reference compound performed in 480 $l$ reactor (●) and 1080 $l$ reactor (○).

The rate coefficient ratios $k_{Epoxide}/k_{Reference}$ obtained from linear regression analyses of these plots are summarized in Table 2. The rate coefficients are averages from the experiments with each reference compound from the experiments performed in both reactors ($cm^3$ $molecule^{-1}$ $s^{-1}$): $k_1$(OH + CHO) = (5.93±1.78) × $10^{-12}$, $k_2$(OH + EHX) = (5.77±1.29) × $10^{-12}$, $k_3$(OH +12EB) = (1.98±0.39) × $10^{-12}$, $k_4$(OH + $c$EB) = (1.50±0.26) × $10^{-12}$, $k_5$(OH + $t$EB) = (1.81±0.42) × $10^{-12}$.

All the rate coefficients obtained in this study, except those for 12EB and CHO, were measured for the first time. The errors presented in Table 2 for the individual value of the slope $k_{Epoxide}/k_{Reference}$ represent two times the standard deviation of the least-squares fit of the data. The error of the slope is the most representative for the quality of the data from a relative kinetic study since the reference rate coefficient could be later reconsidered once new studies would be performed. The epoxides rate constants are placed on an absolute basis by using different reference rate constants for which a uniform error of 10% has been considered. The final rate constant has been calculated as an average of the individual rate constant with the error calculated using the arithmetic calculation of the error propagation.

**Table 2.** Measured rate coefficient ratios, $k_{Epoxide}/k_{Reference.}$, and the rate coefficients for the reactions of OH radical with epoxides at (298±3) K derived from these ratios.

| Epoxide | Reference compound | $k_{Epoxide}/k_{Reference.}$ | $k_{Epoxide}$ (cm$^3$ molecule$^{-1}$ s$^{-1}$) |
|---------|--------------------|------------------------------|-------------------------------------------------|
| CHO | Propylene | 0.258±0.009 | (6.28±0.66) ×10$^{-12}$ |
|  | Ethylene | 0.751±0.038 | (6.40±0.72) ×10$^{-12}$ |
|  | Isobutene | 0.099±0.005 | (5.13±0.57) ×10$^{-12}$ |
|  | **Average** |  | **(5.93±1.13) ×10$^{-12}$** |
| EHX | Propylene | 0.259±0.004 | (6.33±0.64) ×10$^{-12}$ |
|  | Ethylene | 0.612±0.009 | (5.21±0.53) ×10$^{-12}$ |
|  | **Average** |  | **(5.77±0.83) ×10$^{-12}$** |
| 12EB | *iso*-Propyl acetate | 0.406±0.011 | (1.51±0.16) ×10$^{-12}$ |
|  | Ethylene | 0.286±0.006 | (2.43±0.25) ×10$^{-12}$ |
|  | **Average** |  | **(1.98±0.29) ×10$^{-12}$** |
| *t*EB | *sec*-Butyl acetate | 0.398±0.009 | (2.25±0.23) ×10$^{-12}$ |
|  | Ethylene | 0.190±0.007 | (1.62±0.17) ×10$^{-12}$ |
|  | Propylene | 0.064±0.002 | (1.56±0.16) ×10$^{-12}$ |
|  | **Average** |  | **(1.81±0.33) ×10$^{-12}$** |
| *c*EB | *iso*-Propyl acetate | 0.518±0.012 | (1.93±0.20) ×10$^{-12}$ |
|  | *sec*- Butyl acetate | 0.142±0.001 | (0.80±0.08) ×10$^{-12}$ |
|  | Butane | 0.745±0.022 | (1.77±0.18) ×10$^{-12}$ |
|  | **Average** |  | **(1.50±0.28) ×10$^{-12}$** |

*3.1 Correlation between the rate coefficients of the reaction of epoxides with OH radicals and chlorine atoms*

The abstraction route defines the main reaction pathway of the epoxides with OH radicals. For epoxides series it is likely to define a correlation of the rate coefficients from their reaction with OH radicals and chlorine atoms. Similar correlations have been observed for the series of alkanes, saturated alcohols and acyclic ethers (Calvert et al., 2011). Figure 3 presents a log-log correlation plot between the Cl atoms and OH radicals rate coefficients with epoxides for the series of propylene oxide, 1,2-epoxyhexane, 1,2-epoxybutane and both isomers of 2,3-epoxybutane. A very clear correlation ($R^2$=0.9935) described by the relation $\log_{10}[k_{(Cl + epoxides)}] = 0.718 \times \log_{10}[k_{(OH + epoxides)}] - 1.685$ has been obtained. The ethylene oxide and cyclohexene oxide has been represented in the Figure 3 but have been not included into the correlation plot, although including the cyclohexene oxide, the log-log correlation would change to $\log_{10}[k_{(Cl + epoxides)}] = 0.788 \times \log_{10}[k_{(OH + epoxides)}] - 0.852$ with ($R^2$=0.978). The reactivities of the epoxides with OH radicals and Cl atoms are clearly correlated for the series of epoxides. However, the log-log correlation for epoxides is different to the one presented by Calvert et al. (2011) described by the relation $\log_{10}[k_{(Cl + alkanes)}] = 0.521 \times \log_{10}[k_{(OH + alkanes)}] - 3.670$ with ($R^2$=0.85) for the series of alkanes with those two atmospheric oxidants. Besides, the log-log correlation for the series of ethers and alcohols with these two oxidants presented by (Calvert et

al., 2011) described by the relation $\log_{10}[k_{(Cl + ether/alcohol)}] = 0.634 \times \log_{10}[k_{(OH+ ether/alcohol)}] - 2.710$ with ($R^2 = 0.72$) is in better agreement with those obtained in this study for epoxides.

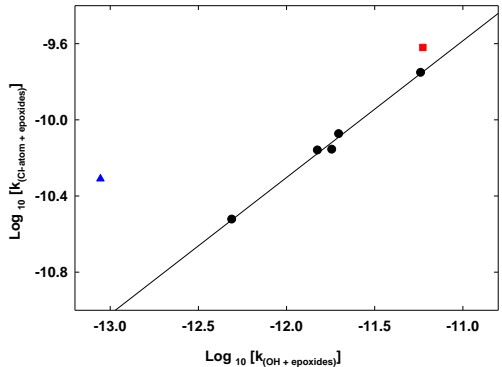

**Figure 3.** Double logarithmic plot ($\log_{10}$) of the rate coefficients for the reaction of Cl-atoms versus the reaction of OH-radicals with the epoxides (propylene oxide, 1,2-epoxybutane, *trans*-2,3-epoxybutane, *cis*-2,3-epoxybutane, 1,2-epoxyhexane). The solid line represents the unweighted least-squares fit to the data. Cyclohexene oxide is represented by the red square while ethylene oxide is represented by a blue triangle. Both were not included into the least squares fit.

The log-log correlation plot can be used to predict rate coefficients. For example, the rate coefficient for the reaction of propylene oxide with chlorine atoms which has not been measured to date, can be derived based on the proposed correlation, $k_{(propylene\ oxide + Cl)}) = 3.04 \times 10^{-11}$ cm$^3$ molecule$^{-1}$ s$^{-1}$. However, the knowledge of rate coefficients for the reactions of epoxides with atmospheric oxidants is still very limited.

*3.2 Structure-Activity relationship (SAR) calculations for epoxides*

To date, the rate coefficients of hundreds of new compounds have been studied under laboratory conditions with different atmospheric oxidants. However, there are still many compounds for which the database is still very scarce, such as for cyclic ethers.

There are several methodologies to estimate the rate coefficients towards OH radicals. One of them is quantum theoretical calculations useful for reactions, which are very difficult to study directly in the laboratory. Another approach is the Structure-Activity Relationship (SAR) method, which rely solely on the structure of the organic compounds and the effect of this structure on the reactivity. Both methodologies are critical to predict the chemical reactivity and physical behaviour of compounds where direct data are not available. The SAR estimation method applied for the epoxide type compounds studied in the present work treats the rate of abstraction from a group within an epoxide molecule. In the case of H atom abstraction from C-H bonds, the calculation of the overall H atom abstraction rate constants is based upon the estimation of -CH$_3$, -CH$_2$-, >CH-, group rate constants.

The -CH$_3$, -CH$_2$-, and >CH- group rate constants depend on the substituents around those groups.

For example, in the case of an α position: $k(CH_3-X) = k_{prim} F(X)$; $k(X-CH_2-Y) = k_{sec} F(X) F(Y)$; and $k(X-CH(Y) (Z)) = k_{tert}$ F(X) F(Y) F (Z). Where $k_{prim}$, $k_{sec}$ and $k_{tert}$ are the rate constants per -CH$_3$, -CH$_2$-, and >CH- group for a "standard" substituent,

X, Y, and Z are the substituent groups; and F(X), F(Y), and F(Z) are the corresponding substituent factors (Atkinson, 1987, 1986a, b). The validity and usefulness of a SAR depends on many factors, such as the amount and type of input data, reliability of prediction, scope of applicability, and ease of implementation (Vereecken et al., 2018).

The rate constants of the reactions with OH radicals obtained in this work were used to derive reactivity trends towards the OH radical. Also, five different approaches have been used in the SAR estimations in order to evaluate the most suitable

method to predict the reactivity of this series of epoxides towards OH radicals. The results obtained are described in more detail below.

For isomers it has been observed that within the experimental error limits, the rate coefficients for the reaction of *trans*-2,3-epoxy butane with OH radicals are slightly different than that for *cis*-2,3-epoxybutane. Table 3 shows the result of the SAR calculations for the epoxide reactions by using different factors from the literature for the total rate constant estimation. Method

(a) Kwok and Atkinson (1995) used the substituent factor F(-OR)=6 to describe the effect of one or two α-ether linkages, with the influence of a β-ether linkage. Method (b) Calvert (2011) also took into consideration the influence of a β-ether linkage and a factor F(-OR)=4.13. Method (c) Calvert (2015) is similar to method (a) and (b) but assuming a factor F(-OR)=8.4. Comparing these approaches with the experimental results, for 1,2-epoxybutane the estimated value using the method (a) Kwok and Atkinson (1995) is the closest to the experimental value. However, it is not the case for cis / trans isomers where a

small deviation is observed.

The SAR estimation (d) Middala (2011) proposed a corrected factor for ring strain, assuming that the effect of the presence of oxygen on the ring might have been underestimated. However, as shown in Table 3, the experimental data obtained in this work differ in most of the cases from the values predicted by SAR using this approach.

The SAR estimation using (e) Jenkin (2018) propose an improved new set of three rate coefficients for H atom abstraction for

C atoms adjacent to ether linkages. These new sets of the rate coefficients are applied independently of neighbouring group substituent factors. Our experimental results show that the rate coefficient of 1, 2-epoxybutane is lower than the rate coefficient calculated with this approach. The rate coefficient calculated for cyclohexene oxide with this approach is the lowest with respect to all experimental values and by other SAR estimation methods.

The influence of the ether group seems not be the same for epoxides of the same number of carbon atoms with differences in

the symmetry. Factors like geometrical disposition and neighbour groups could explain the H atom abstraction in places far from the ether linkage.

In less substituted epoxides, such as propylene oxide the approaches presented by Kwok and Atkinson (1995), Calvert (2011), and Calvert (2015) are the closest to the experimental values determined in this work.

There are discrepancies between the values calculated by different approaches of the SAR estimation method, as is shown in

Table 3. In addition, there is no SAR estimation method that consistently generates a value close to the experimental value for all compounds, which have been studied in the present work. These discrepancies had already been observed since the first

SAR approximations for cyclic ethers and they could be attributed to the limited database that still exists for this kind of compounds (Kwok and Atkinson, 1995; Jenkin et al., 2018). In general, all these approaches have been developed to fit the behaviour of cyclic alkanes, with one factor being attributed to ring strain and the presence of an ether group.

**Table 3.** Comparison of the rate coefficients for the reaction of OH radicals with the epoxides investigated in this study with the literature data and SAR predictions $k_{Epoxide}/k_{SAR}$

| Epoxide | $k_{SAR}$ | $k_{Epoxide}/k_{SAR}$ | $k_{Epoxide}$ $(cm^3\ molecule^{-1}\ s^{-1})$ | Reference |
|---|---|---|---|---|
| Cyclohexene Oxide | [a]$6.37\times10^{-12}$ | 0.93 | $(5.93\pm1.13)\times10^{-12}$ | This work |
|  | [b]$6.14\times10^{-12}$ | 0.97 | $(6.51\pm0.65)\times10^{-12}$ | (El Othmani et al., 2021a) |
|  | [c]$6.31\times10^{-12}$ | 0.94 |  |  |
|  | [d]$5.81\times10^{-12}$ | 1.02 |  |  |
|  | [e]$1.02\times10^{-11}$ | 0.58 |  |  |
| Ethylene Oxide | [a]$2.80\times10^{-13}$ | 0.31 | $(0.88\pm0.25)\times10^{-13}$ | (Calvert et al., 2011) |
|  | [b]$1.89\times10^{-13}$ | 0.47 |  |  |
|  | [c]$3.54\times10^{-13}$ | 0.25 |  |  |
|  | [d]$0.77\times10^{-13}$ | 1.14 |  |  |
|  | [e]$8.85\times10^{-14}$ | 0.99 |  |  |
| Propylene Oxide | [a]$0.59\times10^{-12}$ | 0.83 | $(0.49\pm0.52)\times10^{-12}$ | (Wallington et al., 1988b) |
|  | [b]$0.46\times10^{-12}$ | 1.07 | $(0.30\pm1.00)\times10^{-12}$ | (Middala et al., 2011) |
|  | [c]$0.67\times10^{-12}$ | 0.73 | $(0.47\pm0.24)\times10^{-12}$ | (Virmani et al., 2020) |
|  | [d]$0.28\times10^{-12}$ | 1.75 |  |  |
|  | [e]$5.43\times10^{-12}$ | 0.90 |  |  |
| 1,2-epoxyhexane | [a]$4.64\times10^{-12}$ | 1.24 | $(5.77\pm0.83)\times10^{-12}$ | This work |
|  | [b]$4.48\times10^{-12}$ | 1.29 |  |  |
|  | [c]$4.15\times10^{-12}$ | 1.39 |  |  |
|  | [d]$3.97\times10^{-12}$ | 1.45 |  |  |
|  | [e]$6.34\times10^{-12}$ | 0.91 |  |  |
| 1,2-epoxybutane | [a]$1.81\times10^{-12}$ | 1.09 | $(1.98\pm0.29)\times10^{-12}$ | This work |
|  | [b]$1.65\times10^{-12}$ | 1.20 | $(1.90\pm0.67)\times10^{-12}$ | (Wallington et al., 1988a) |
|  | [c]$1.83\times10^{-12}$ | 1.08 | $(2.20\pm0.02)\times10^{-12}$ | (El Othmani et al., 2021b) |
|  | [d]$1.45\times10^{-12}$ | 1.37 |  |  |
|  | [e]$2.96\times10^{-12}$ | 0.67 |  |  |
| *trans*-2,3-epoxybutane | [a]$0.92\times10^{-12}$ | 1.97 | $(1.81\pm0.33)\times10^{-12}$ | This work |
|  | [b]$0.73\times10^{-12}$ | 2.48 |  |  |
|  | [c]$0.99\times10^{-12}$ | 1.83 |  |  |
|  | [d]$0.49\times10^{-12}$ | 3.69 |  |  |
|  | [e]$9.98\times10^{-13}$ | 1.81 |  |  |
| *cis*-2,3-epoxybutane | [a]$0.92\times10^{-12}$ | 1.63 | $(1.50\pm0.28)\times10^{-12}$ | This work |
|  | [b]$0.73\times10^{-12}$ | 2.05 |  |  |
|  | [c]$0.99\times10^{-12}$ | 1.52 |  |  |
|  | [d]$0.49\times10^{-12}$ | 3.06 |  |  |
|  | [e]$9.98\times10^{-13}$ | 1.50 |  |  |

Calculated using the SAR estimation of (a) (Kwok and Atkinson, 1995), (b) (Calvert et al., 2011), (c) (Calvert et al., 2015), (d) (Middala et al., 2011), (e) (Jenkin et al., 2018).

### 3.2.1 Pseudo-ethylenic character in the epoxy ring as main factor affecting SAR estimation

The kinetics, reactivity trends and thermochemistry of cis/trans-2,3-epoxybutane, 1,2-epoxybutane, 1,2-epoxyhexane and cyclohexene oxide initiated by Cl atoms were previously discussed (Tovar et al., 2021). The series of epoxides were theoretically investigated using density functional theory (DFT) applying the B3-LYP exchange-correlation function with Grimme´s GD3-BJ empirical dispersion corrections. The calculations results showed that the abstraction channels of $\beta$-$CH_2$ sites are dominant for all epoxides under study.

As we explained in Tovar et al. (2021), the sites in the molecule are more reactive in $\beta$-position when the carbon atoms in ethylene oxide approach the $sp^2$ state and its $CH_2$ units are pseudo-ethylenic in character. That conjugation does not occur if the $\pi$ orbitals of the unsaturated groups cannot orient themselves with their axes parallel to the plane of the epoxide ring. Such a geometric arrangement in the epoxy molecule would be necessary to allow the abstraction of a hydrogen in the molecule in a preferential way, thus increasing the reactivity of the reaction.

In the section 3.2 we discuss the rate coefficients experimentally calculated for the same series of epoxides towards OH radical, and the estimated rate constants of epoxides with total of two to six carbon number by using different SAR approaches. In these estimations different substituent factors were applied. In the following discussion, we present the possible theoretical reasons why these SAR estimations could differ between them and with respect to the experimental values based in novel theoretical and experimental studies that confirms our previous findings.

From Table 4 we can assess the influence of the ring size over a series of cyclic ethers. In this way, the cyclic ether reactivities towards OH radicals are increasing with the ring size (k in $10^{-12}$ cm$^3$ molecule$^{-1}$ s$^{-1}$) ($k_{oxirane(C2H4O)} = (0.091 \pm 0.023)$) < ($k_{oxetane(C3H6O)} = (10.3 \pm 2.6)$) < ($k_{oxolane(C4H8O)} = (17.0 \pm 2.6)$); ($k_{oxaneC5H10O)} = (12.0 \pm 2.0)$) < ($k_{oxepane(C6H12O)} = (18.0 \pm 5.0)$).

For previous studies we know that the electron ability in hydrogen bonding can act as a measure of the relative electron density (Tamres et al., 1954). The very low electron donor ability of substituted ethylene oxides, compared with other cyclic ethers, in hydrogen bonding has already been shown in the past (Searles and Tamres, 1951; Searles et al., 1953). Thus, the oxygen in 3-membered ring ethers is more electronegative than in larger ring ethers or in acyclic ethers. This observation can be explained from Walsh´s proposal for the structure of ethylene oxide and cyclopropane (Walsh, 1949). Walsh's model imply a high electron density in the centre of the three-membered rings, which suggests that the exterior of such rings would have a lower electron density than normal for the atoms involved. This effect is explained for strained systems where the hybridized orbitals employed for a bonding attain maximum overlap in such a way that altered valence angles "bent bonds" are realized. Because the Walsh model does not correspond to the ground state of cyclopropane/epoxide, the description of bent bonds for cyclopropane by the Förster-Coulson-Moffitt model (Förster, 1939; Coulson and Moffitt, 1949, 1947) is the most frequently considered one (Wiberg, 1996). More recently, the topological analysis of the electron density method (ED) is applied to describe the electron distribution within a compound (Kutzelnigg, 1993; Bader et al., 1994; Coppens, 2005; Koritsanszky and Coppens, 2001). However, the ED method provides information on concentration and depletion of electrons, but not on the pairing of electrons. In order to measure the electron pair localization the electron ability indicator (ELI) it is used (Kohout, 2004). A recent study presented by Grabowsky (2010), has introduced the ELI based on an X-ray diffraction experiment by

means of the X-ray constrained wavefunction fitting procedure. The method was applied on a series of epoxides derivatives, and clearly indicated outwardly bent bonds according to the Förster-Coulson Moffit model.

Also, this study shows that the maxima of deformation density and the valence shell charge concentrations (VSCCs) in the Laplacian maps are located clearly outside the bond axes for both the C-O and C-C bond, and there are no maxima inside the ring. The Förster-Coulson Moffit model makes use of hybrid orbitals with a relation between s- and p- character like sp2-hybridization. This conducts to an orbital overlapping outside the direct bond axis forming three σ-type bonds. The orbitals used in these exterior bent bonds of the ring are in a favourable position for some overlap with p-orbitals from adjacent atoms (Searles et al., 1953). The involvement of $sp^2$-type orbitals in the epoxide ring instead of $sp^3$ like in normal single bond suggests that the bonds are not saturated and can interact with π-electron systems (Grabowsky et al., 2010). Our recent study on the reactivity of a series of epoxides towards chlorine atoms in gas phase using quantum mechanical calculations agrees with these findings. As we discussed in Tovar et al. (2021), If we consider the case of cis/trans-epoxybutane, when the compound has lost one H of the $CH_3$ group, it becomes trigonally ($sp^2$) hybridized, facilitating the possible formation of a double bond with the singly occupied p-orbital after abstraction of H. This effect could explain the formation of acrolein from Cl-initiated oxidation of both *cis*-2,3-dimethyloxirane and *trans*-2,3-dimethyloxirane with yields of 36.49±0.55 and 49.08±1.29 at 650K and 800k, respectively in the work presented by Doner et al. (2021). The formation of the vinyl oxirane from beta hydrogen abstraction and the subsequent formation of the keto hydro peroxy radical reported by Christianson et al. (2021) for the oxidation of 1,2-epoxybutane also confirms this theory. In addition, one of the most abundant products reported in the same work, 2,2-bioxirane (31.44±6.49) at 640K, would also respond to the same reactivity trend, with abstraction in the beta position being the most predominant.

This structural effect has been observed also experimentally with cyclopropyl ketones (Rogers, 1947), α,β-epoxyketones (Walsh, 1949; Cromwell and Graff, 1952), ethylenimine ketones (Cromwell and Graff, 1952) and more recently in the acid catalysed ring opening of epoxides (Oshima et al., 2008). Moreover, there is some evidence of a conjugation of the epoxy ring with substituents using methods like UV spectroscopy, heat of combustion and MO calculations (Parker and Isaacs, 1959; Starit et al., 1964).

*3.2.2 Ring strain in epoxides*

From Table 4 we can see that the epoxides are less reactive towards OH radicals than the analogous alkanes with the same number of carbon atoms: (k in $10^{-12}$ $cm^3$ molecule$^{-1}$ s$^{-1}$) ($k_{cEB(C4H8O)}$ = (1.50±0.28)), ($k_{tEB(C4H8O)}$ = (1.81±0.33)), ($k_{12EB(C4H8O)}$ = (1.98±0.29)) < ($k_{butane(C4H10)}$ = (2.38±0.24)). The effect is more pronounced for epoxides with 4 carbon atoms than for those with 6 carbon atoms: ($k_{12EHX (C6H12O)}$ = (5.77±0.83)) < ($k_{(C6H14)}$ = (5.86±1,17)). The presence of a six-membered ring next to the epoxide group increases its reactivity ($k_{CHO(C6H10O)}$ = (5.93±1.13))> ($k_{(C6H14)}$ = (5.86±1,17)).

Grabowsky et al. (2010) derived the experimental electron density of ethylene oxide from a multipole refinement of 100 K X-ray data and complemented by density-functional calculations at experimental and optimized geometry. This study found that despite the high strain in the three-membered ring of ethylene oxide, most atomic and bond topological properties do not differ

from comparable fragments in unstrained molecules. Grabowsky et al. (2010), concluded that the strained and unsaturated character of the epoxide ring is reflected in the populations of the bonds.

The three-membered ring strain is considered in the SAR estimations for its influence on the reactivity of epoxides. However, this factor itself does not play a dominant role for the rate coefficients of the epoxides. The presence of the epoxy ring would affect the reactivity of epoxides because the CH group attached favours the pi overlap (Figure 4). For this reason, the factors

(F(O), F(3m-ring)) need to be treated specially in SAR and possibly as one single factor F (epoxy ring-β-$C_X$) since their influence on the beta carbon is very pronounced. Only then, see the contributions of other groups, linear, branched, rings etc. on this basic structure. The reactivity trends observed in our previous (Tovar et al., 2021) and present work suggests that the rate constant of the β-channel could be the closest to the $k_{total}$ determining the overall reaction rate. These aspects had not been considered in SAR estimates to date, therefore new factors that includes these complicated bond structures in epoxides need

to be developed.

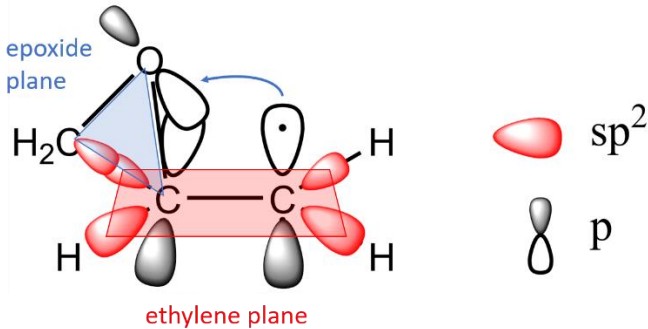

**Figure 4**. Involvement of sp$^2$-type orbitals in the epoxide ring

*3.3 Improving SAR estimation for epoxides*

In order to provide a new set of "corrected" factors, especially for F (epoxy ring-β-$C_X$) that can be used for an improved SAR estimation for epoxides a new set of theoretical calculations with OH radicals are recommended. This kind of calculations are beyond of the scope of this work. However, a possible approach should conduct to expanding the kinetic database of epoxides (experimental and theoretical) in a wide temperature range. Experiment and theory are highly complementary in this regard, and their combined application allows for alleviation of some of the shortcomings in both approaches; for theoretical work

specifically, even a single experimental datum point often allows for the optimization of the PES or energy transfer parameters to allow T, P-extrapolations with strongly enhanced reliability (Vereecken et al., 2015).

A recent work by Wang et al. (2022) proposed a concept of experiment-SAR-DFT combined to theoretically discuss the abstraction reaction of a series of alcohols and ketones. Similar approach could be used for epoxides with some considerations. As has been shown in Tovar et al. (2021), the potential energy surface of chlorine atoms reacting with the series of epoxides

possess barrierless reaction. For these cases, Piletic et al. (2017) proposed to use the Master Equation Solver for Multi-Energy

Reactions (MESMER v.4.0) in order to compute the energy dependent microcanonical rate constants for all forward and reverse reactions on the PES with the option of computing the k(E)s using an inverse Laplace transform (ILT) technique. The k(E)s may be calculated by using Arrhenius parameters from a fit of the temperature dependent high pressure rate coefficients. A second method could include a semi-empirical method using the Arrhenius parameters in MESMER to fit phenomenological

rate constants that are consistent with the experimentally measured rates. Our preliminary exploration of the PES of the series of epoxides towards chlorine atoms Tovar et al. (2021) provide a hint of the complex structural analysis that must be developed in order to assure more accurate rate coefficients and capture the reactivity trends of epoxides by SAR estimation method. Some studies have evaluated the temperature dependence for 1,2- epoxypropane and 1,2-epoxybutane towards OH radicals. Virmani et al. (2020) found for the reaction with 1,2-propylene oxide a very small (or possibly no) dependence on temperature

within the temperature range between 261-355 K. El Othmani et al. (2021b) showed for the reaction with 1,2-epoxybutane a very weak negative temperature dependence at T≤285 K and a transition toward a positive temperature dependence at T≥295K. (El Othmani et al. (2021a) measured a stronger negative temperature dependence in the case of cyclohexene oxide. This behaviour could suggest the existence of van der Waals complexes that can play a role in the reaction mechanisms of epoxides at lower temperatures. More theoretical research is needed to further elucidate these findings.

All the electronic structure calculations will serve as input for the kinetic rate calculations using MESMER o another similar approach.

Even with the clear correlation between the rate coefficients of epoxides with OH radicals and chlorine atoms showed in section 3.1, a detailed structural optimizations and vibrational frequency calculations are also recommended for the reactions of the epoxides towards OH radicals.

Parallel to the present work, product studies of the reactions presented here have been undertaken, to delve a little deeper into the reaction mechanisms of these reactions.



Table 4. Reactivity trends of the epoxy compounds towards OH radicals compared with their homologous alkanes, acyclic ethers and cycloalkanes with the same number of carbon atoms.

| | Cyclic ether | | | Alkane/cycloalkane | | | Acyclic ether | |
|---|---|---|---|---|---|---|---|---|
| Ring size | Structure | $k_{OH}\times10^{12}$ † | C Atoms | Compound | $k_{OH}\times10^{12}$ † | C atoms | Compound | $k_{OH}\times10^{12}$ † |
| 7 member | oxepane | 18.0[1] | | | | | | |
| 6 member | oxane | 12.0[1] | 7 | methylcyclohexane[1]<br>heptane[1] | 9.26[1]<br>6.23[1] | | | |
| 5 member | oxolane | 17.0[1] | 6 | 2,3-dimethylbutane<br>Hexane | 5,86[1]<br>4.97[1] | 6 | di-n-propyl ether | 20.0[1] |
| 4 member | oxetane | 10.3[1] | 5 | 2-methylbutane<br>Pentane | 3.74[1]<br>3.76[1] | 5 | methyl-s-butyl-ether | 9.67[1] |
| 3 member | CHO 5.93<br>EHX 5.77<br>12EB 1.98<br>tEB 1.81<br>cEB 1.50<br>oxirane 0,09 | | 4 | butane | 2.38[1] | 4 | 1-metoxypropane | 9.91[1] |

† units: cm$^3$molecule$^{-1}$s$^{-1}$

[1](McGillen et al., 2020); (Recomended values of the most recent revision of database for the kinetics of the gas-phase atmospheric reactions of organic compounds)

## 4 Atmospheric implications

Using the kinetic data obtained in the present work, in combination with daytime average radical concentrations, the estimation of the tropospheric lifetimes of the investigated compounds can be calculated using the expression:

$$\tau_{OH/Cl} = 1/k_{OH/Cl}[OH/Cl]$$

Therefore, for a globally averaged [OH] of $11.3 \times 10^5$ radicals cm$^{-3}$ (Lelieveld et al., 2016) the atmospheric lifetimes for cyclohexene oxide, 1,2-epoxyhexane, 1,2-epoxybutane, *trans*-2,3-epoxybutane and *cis*-2,3-epoxybutane are estimated between 1 to 7 days, as shown in Table 5.

According to the rate coefficients presented in Tovar et al. (2021) and considering an estimated atmospheric concentration [Cl] of $2.3 \times 10^3$ atoms cm$^{-3}$ (Young et al., 2014), an atmospheric lifetime between 21 and 73 days has been calculated for the epoxides. Accordingly, the reaction with OH radicals is the primary sink for epoxides in the atmosphere under these conditions. However, in some cases the atmospheric composition can be more than an order of magnitude more reactive with Cl than with OH (e.g., Los Angeles during CalNex where a maximum [Cl] of $8.7 \times 10^3$ atoms were measured (Young et al., 2014)). In polluted continents during winter ClNO$_2$ chemistry drives increases in ozone of up to 8 ppb (Wang et al., 2019).

**Table 5.** Comparison of the atmospheric lifetimes of epoxides towards OH radicals and chlorine atoms.

| Epoxides | $\tau_{OH}$ (days) | $\tau_{Cl}$ (days) |
|---|---|---|
| cyclohexene oxide | 1.7 | 20.9* |
| 1,2-epoxyhexane | 1.8 | 28.4* |
| 1,2- epoxybutane | 5.1 | 59.7* |
| trans-2,3-epoxybutane | 5.7 | 71.9* |
| cis-2,3-epoxybutane | 6.8 | 72.7* |

* Calculated values using the rate coefficients of epoxides towards Cl atoms (**Tovar et al., 2021**).

In the laboratory and ambient SOA, epoxides are considered as potential precursors for sulphate esters, polyols, hydroxy nitrates and halides. Epoxide concentrations in the aerosol composition will be limited by the rates of the gas uptake process, which will be largely determined by the gas phase epoxide concentration levels (Minerath and Elrod, 2009; Minerath et al., 2009), the uptake process and particles acidity. Higher acidity of aerosol particles would produce more efficient heterogeneous reactions of epoxides. A recent study suggests that heterogeneous reaction of some epoxides on acidic aerosols are faster and more efficient than gaseous reactions with atmospheric oxidants, and thus these reactions could represent the major removal

pathway for epoxides (Lal et al., 2012). Besides, in urban environments the particle acidity can be significantly high, with pH between 0-5. Under such conditions, epoxides can react rapidly, and the reaction products of the heterogeneous reactions could
contribute to the growth of the SOA particles and to the possible modification of their physical and chemical properties (Zhang et al., 2007). Minerath and Elrod (2009) have shown that 1,2-epoxybutane, which possess slower hydrolysis, has an increasing lifetime with increasing pH. It is expected that even at higher pH most of the epoxides will have time to react on SOAs. The monitoring and evaluating of emissions of this kind of compounds in different ambient conditions, as pH and different regimens of $NO_x$, is recommended in rural and urban locations.
Monitoring and evaluating the emissions of these compounds under different ambient conditions, as pH and different regimens of $NO_x$, is recommended in rural and urban locations.

**Conclusions**

The rate coefficients for the reaction of OH radicals with three different epoxides have been determined for the first time in this study. The rate coefficient for 1,2-epoxy butane and cyclohexene oxide are in very good agreement with previous studies
(Wallington et al., 1988a; El Othmani et al., 2021b, a). A comparison of the reactivity trends from the gas-phase reaction of the epoxides obtained in this study showed a very good correlation with the reactivity trends of epoxides with chlorine atoms. Differences have been observed between the experimental OH rate coefficients and those obtained from SAR estimations. However, the values determined by the SAR method show some discrepancies dependent on the substituent factors, which were considered for the calculations. Such discrepancies could be explained by taking into consideration the structural effect
of the ring acting in conjunction with the oxygen atom and the unsaturated character of the epoxy ring. From the atmospheric chemistry viewpoint, the rate coefficient from the present study would help to extend the database for the reaction of cyclic ethers under atmospheric conditions.

*Data availability*. Data can be provided upon request to the corresponding author Carmen Maria Tovar
carmen.tovar_ramos@uni-wuppertal.de

*Author contributions*. CMT conducted the experiments. CMT performed the data analysis. IB, IGB, and PW made revisions at different stages of the study. CMT prepared the paper with contributions from all co-authors.

*Competing interests*. The contact author has declared that neither they nor their co-authors have any competing interests.

## Acknowledgement

The authors acknowledge the financial support provided by the European Union's Horizon 2020 research and innovation programme, through the EUROCHAMP-2020 Infrastructure Activity Grant (grant agreement no. 730997). IGB acknowledges the PN-III-P4-ID-PCE-2016-4-0807 and PN-III-P2-2.1-PED-2019-4972 UEFISCDI projects. CMT is grateful for a PhD scholarship granted by FANTEL.

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
