# Peer review of "Kinetic Study of the Atmospheric Oxidation of a Series of Epoxy Compounds by OH Radicals"

_Atmospheric Chemistry and Physics, 2021_

## Author Comment (AC1)

**Response to the reviewer's comments for manuscript acp-2021-962**

**Title: "Kinetic Study of the Atmospheric Oxidation of a Series of Epoxy Compounds by OH Radicals"**

by Carmen Maria Tovar et al., Atmos. Chem. Phys. Discuss., https://doi.org/10.5194/acp-2021-962-RC1, 2022

Special Issue: Simulation chambers as tools in atmospheric research (AMT/ACP/GMD inter-journal SI)

We are very grateful for the valuable suggestions and comments of both reviewers, which were carefully considered and helped to improve the quality of our manuscript. The original comments from the reviewers are shown in black. Our answers are marked in blue while changes performed in the body of the manuscript are marked in red.

**A. Comments by Referee #1**

**R1:** *Kinetic Study of the Atmospheric Oxidation of a Series of Epoxy Compounds by OH Radicals* by Carmen Maria Tovar et al. presents a series of measurements of epoxides using the relative rate technique with a variety of reference compounds. This work fills in gaps in the kinetic information on these compounds that are used in industrial chemistry and potentially  $CO_2$  capture technology. This work also discusses the failings of the current structure activity relationships (SARs) used to estimate the rate constants of epoxides with OH radicals. This work adequately presents their laboratory results and comparison to the SARs, however it is unclear what actually needs to be adjusted to the current SARs from the presented results, as there was no attempt to create a more representative SAR. Because of the additional knowledge presented in this paper on the rate constants of these chemicals and potential to improve current SARs for epoxides, I recommend that this is reconsidered for publication after major revisions.

**Reply from the authors:** We gratefully appreciate the referee's comments. Our work has been indeed meant to contribute with additional rate coefficients to improve our current knowledge about the gas-phase reactivity of epoxides toward OH radicals. Kinetic studies with OH radicals on this group of compounds, widely spread in the atmosphere, have been limited to ethylene oxide and propylene oxide and just very recently the kinetic database has been updated with single studies of additional kinetic rate constants.

R1: Major Comments: L285-327: "(Tamres et al., 1954) stated...means of theoretical calculations." This seems beyond the scope of the paper. I think that it is useful to think of the reactivity of these compounds from a theoretical perspective in some situations. However, structure activity relationships are observationally/experimentally derived. In these paragraphs, there are many facts stated about the electronegativity, ring strain, hybridization, and other parameters of epoxides, but it is unclear how these relate to your results. Are these factors that can be put into a revised SAR? Can you obtain these from just knowing the structure of the molecule? As a concrete example, how does the suggestion "that the bonds are not saturated and can interact with  $\pi$ -electron systems" affect the OH rate constants with the epoxides? If you keep this section, it should be connected to the SAR and reaction rates more clearly.

**Reply from the authors:**

We agree with the reviewer that the theoretical basis and the experimental findings should be better linked in the text and decide to rewrite the section. The obtained rate coefficients from the present study together with existing literature data do not provide enough information to improve current SAR methods and more data are required to properly capture the behaviour of the epoxide reactions with OH radicals. However, the discussion section has been improved organizing the information about the crucial factors in the improvement of a SAR estimation method for future work. Also, we include some recommendations since this are a common observation for both authors. We also add equations based on the log-log correlation fitting data between the kinetic rate coefficients for the reactions with OH and chlorine atoms and an extra figure to enrich the discussion.

**Changes in the manuscript/supplementary text:**

**L235: 3.1 Correlation of the rate coefficients between the reaction of epoxides with OH radicals and chlorine atoms**

[revised manuscript text omitted]

**R1:** Table 1: What is the error of your concentrations? I don't think the error on the 1ppm starting concentration is the same as the 84ppm, which is what this table implies. Were the initial concentrations assumed from the known amount injected? If so, how do you know it all made it into the chambers without decaying? How did you actually manage to get these chemicals into the gas-phase consistently?

**Reply from the authors:** We agree that usually the concentrations (mixing ratios in Table 1) should be represented by a value with a range of uncertainties. A statement has been introduced in the text to show how this concentration has been obtained. However, we decided, deliberately, not to provide uncertainties for the concentrations since they are not required for the experimental approach applied in our study. The exact concentrations are not representative for a relative rate kinetic measurement and are not included in any calculations since we follow in our evaluations the subtraction factor of a reference IR spectrum from all IR spectra recorded during a kinetic experiment. For equation (I) there is a direct dependence between the ratios of concentrations with the ratio of subtraction factors and, accordingly, such ratios are considered in the relative rate coefficient determinations.

From Table 1 the highest initial mixing ratios (ppmv) for both reactors are representative for the precursor of OH radicals (i.e. methyl nitrite), which is added to ensure the formation of OH radicals through its photolysis. Usually, methyl nitrite is not completely consumed during the duration of the experiment. Also, we stated (line 144) that a conversion of maximum 50% was achieved through the kinetic reactions for epoxides and reference compounds.

Our previous work on the determination of the kinetic rate coefficients of same group of epoxides with chlorine atoms has been very helpful to handle the different epoxides in the present study.

**Changes in the manuscript/supplementary text:**

**Changes in the manuscript:**

Line 120: The concentrations or mixing ratios were measured from the exact amount of the compound added to the chamber. The liquids and gases have been injected into the chamber via a septum using a gas-tight syringe in a flow of air with the reactor being under reduced pressure. The glass-steel inlet system has been heated up to 60°C to ensure the addition of the entire amount of epoxides and reference compounds into the reactor. After addition and filling the chamber to 760±10 Torr pressure of synthetic air, up to 15 infrared spectra (15 min) have been recorded to prove the constant concentration of the compounds in the chamber. These tests have been performed not only to check if there is a diffusion of the injected compounds in order to correct the rate coefficients for additional loss processes. Any diffusion from the inlet would affect the linearity of the kinetic plots, which otherwise would not have been obtained. The different epoxides and reference compounds employed in the present study did not show a wall loss during the time of the experiment.

Line 160: **Table 1.** Initial mixing ratios used in the 1080 I reactor and 480 I reactor for the epoxide and reference compounds in ppmv (1 ppm =  $2.46 \times 10^{13}$  molecule cm-3) at 298 K and 760 Torr of total pressure. The mixing ratios have been estimated from injections of the compounds into the reaction chambers and provide information about the amount of compounds used in this study.

|                                               | Initial mixing ratios (ppmv) |                |  |  |  |
|-----------------------------------------------|------------------------------|----------------|--|--|--|
| Compounds                                     | Reactor (1080 L)             | Reactor (480L) |  |  |  |
| Epoxides                                      |                              |                |  |  |  |
| cyclohexene oxide                             | 6                            | 17             |  |  |  |
| 1,2-epoxyhexane                               | 6                            | 28             |  |  |  |
| 1,2-epoxybutane                               | 6                            | 23             |  |  |  |
| cis-2,3-epoxybutane                           | 3                            | 28             |  |  |  |
| trans-2,3-epoxybutane                         | 3                            | 23             |  |  |  |
| Reference compounds                           |                              |                |  |  |  |
| Propylene                                     | 4                            | 8              |  |  |  |
| Butane                                        | 4                            | -              |  |  |  |
| iso-propyl acetate                            | 1                            | 9              |  |  |  |
| sec -butyl acetate                     | 1                            | 9              |  |  |  |
| Ethylene
methyl nitrite (OH radical | 4                            | 21             |  |  |  |
| precursor)                                    | 8                            | 84             |  |  |  |

R1: Table 2: The final errors on the rate constants do not seem to make sense. The error on the Ethylene rate constant is ~1.5% and the error of your measured slope is ~5% (is this just the error on the slope? Do you force the fit through zero? Does that change the error?). How does the reported rate constant error for CHO come out to be ~16%? You say "up to 35% contribution in the recommended values of the rate coefficients for reference compounds" is accounted for in the error, but not all of your reference compounds have this large of errors. Also, unless you have a valid justification, it does not make sense for reported errors to have more than one significant figure. On a similar note, the error should also determine the significant figures of the reported value. This might end up affecting your claim that the trans- and cis- isomers have different rates, since the reported rates are within each other's errors.

**Reply from the authors:**

We did not force the fit through zero since the data plotted in the form of equation "I" should be with almost "zero" intercept and our intention was to show that there is not a significant deviation from "zero". The linearity of the plots indicates the absence of interferences and secondary reactions, which can affect the ratio of the rate coefficients  $k_{\text{Epoxide}}/k_{\text{reference}}$ .

In order to provide an unitary answer regarding the error calculation for all rate constants obtained in the present study, we revised the errors for all measured rate constants. The epoxides rate constants are placed on absolute basis by use of different reference rate constants. Since there are various modes for error calculations for these reference rate coefficients from simple standard deviation to average values from different sources, it is recommended to use a uniform representation of the error. The reviewer is correct and the consideration of "up to 35%" for the reference rate constant does not represent all rate constants. We decided to use a more representative error calculation for the epoxide's rate constants. Accordingly, we have adopted the following procedure and the manuscript has been changed in agreement with new assumptions.

The errors presented in Table 2 for the individual value of the slope  $k_{Epoxide}/k_{reference}$  are 2 times the standard deviation of the least-squares fit of the data. The epoxides rate constants are placed on an absolute basis by using different reference rate constants for which we have considered a uniform error of 10%. The final rate constant has been calculated with the expression:

$$k_{average} = \frac{\sum_{i=1}^{n} k_i}{n}$$

while the errors have been calculated using the arithmetic calculation of the error propagation

$$\Delta k_{\text{average}} = \sqrt{\sum_{i=1}^{n} (\Delta k_i)^2}$$

where  $k_i$  are the individual rate constant obtained using one reference compound and  $\Delta k_i$  and  $\Delta k_{average}$  are the associated errors for  $k_i$  and  $k_{average}$ , respectively.

**Changes in the manuscript/supplementary text:**

**Changes in the manuscript:**

Line 201 All the rate coefficients obtained in this study, except those for 12EB and CHO, were measured for the first time. The errors presented in Table 2 for the individual value of the slope  $k_{Epoxide}/k_{reference}$ .represent two times the standard deviation of the least-squares fit of the data. The error of the slope is the most representative for the quality of the data from a relative kinetic study since the reference rate coefficient could be later reconsidered once new studies would be performed. The epoxides rate constants are placed on an absolute basis by using different reference rate constants for which a uniform error of 10% has been considered. The final rate constant has been calculated as

an average of the individual rate constant with the error calculated using the arithmetic calculation of the error propagation.

| Epoxide | Reference                  | $k_{Epoxide}/k_{Reference.}$ | k Epoxide                               |  |  |
|---------|----------------------------|------------------------------|-----------------------------------------------------------|--|--|
|         | compound                   |                              | (cm 3 molecule -1 s -1 ) |  |  |
| СНО     | Propene                    | 0.258±0.009                  | (6.28±0.66)×10 -12                             |  |  |
|         | Ethene                     | 0.751±0.038                  | (6.40±0.72)×10 -12                             |  |  |
|         | Isobutene                  | 0.099±0.005                  | (5.13±0.57)×10 -12                             |  |  |
|         |                            | Average                      | (5.93±1.13)×10 -12                             |  |  |
|         |                            |                              |                                                           |  |  |
| EHX     | Propene                    | 0.259±0.004                  | (6.33±0.64)×10 -12                             |  |  |
|         | Ethene                     | 0.612±0.009                  | (5.21±0.53)×10 -12                             |  |  |
|         |                            | Average                      | (5.77±0.83)×10 -12                             |  |  |
|         |                            |                              |                                                           |  |  |
| 12EB    | iso-Propyl acetate         | 0.406±0.011                  | (1.51±0.16)×10 -12                             |  |  |
|         | Ethene                     | 0.286±0.006                  | (2.43±0.25)×10 -12                             |  |  |
|         |                            | Average                      | (1.98±0.29)×10 -12                             |  |  |
|         |                            |                              |                                                           |  |  |
| tEB     | sec-Butyl acetate          | 0.398±0.009                  | (2.25±0.23)×10 -12                             |  |  |
|         | Ethene                     | 0.190±0.007                  | (1.62±0.17)×10 -12                             |  |  |
|         | Propene                    | 0.064±0.002                  | (1.56±0.16)×10 -12                             |  |  |
|         |                            | Average                      | (1.81±0.33)×10 -12                             |  |  |
|         |                            |                              |                                                           |  |  |
| сEВ     | iso -Propyl acetate | 0.518±0.012                  | (1.93±0.20)×10 -12                             |  |  |
|         | sec- Butyl acetate         | 0.142±0.001                  | (0.80±0.08)×10 -12                             |  |  |
|         | Butane                     | 0.745±0.022                  | (1.77±0.18)×10 -12                             |  |  |
|         |                            | Average                      | (1.50±0.28)×10 -12                             |  |  |

**Table 2.** Measured rate coefficient ratios,  $k_{Epoxide}/k_{Reference.}$  and the rate coefficients for the reactions of OH radical with epoxides at (298±3) K derived from these ratios.

R1: Table 4 (and corresponding discussion): There should be more discussion about what a homologous compound is. Potentially having the structures drawn would help with this. From my perspective, I just see that CHO is C6H10O, methylcyclohexane is C7H14, and din-propyl ether is C6H14O which have different numbers of total atoms. I expect the rate constant for these other compounds to be faster because there are more hydrogens to abstract, so I am not sure if your point is coming across with this table and discussion. It is difficult to see how the addition of the 3-membered ring or additional oxygen actually affects the rates when the molecules could have other factors changing their rate constants.

**Reply from the authors:**

We agree with the reviewer's suggestions and Table 4 and the corresponding discussions have been changed. Also, we have added the structures drawn to the Table.

**Changes in the manuscript/supplementary text:**

**Changes in the manuscript:** Table 4 - Reactivity trends of the epoxy compounds towards OH radicals compared with their homologous alkanes, acyclic ethers and cycloalkanes with the same number of carbon atoms.

|           | Cyclic ether                                   |                                     |            | Alkane/cycloalkane                              |                                        |            | Acyclic ether            |                                     |
|-----------|------------------------------------------------|-------------------------------------|------------|-------------------------------------------------|----------------------------------------|------------|--------------------------|-------------------------------------|
| Ring size | Structure                                      | k он ×10 12 † | C
atoms | Compound                                        | k он ×10 12 †    | C
atoms | Compound                 | k он ×10 12 † |
| 7 member  | Oxepane                                        | 18.0 1                   |            |                                                 |                                        |            |                          |                                     |
| 6 member  | Oxane                                          | 12.0 1                   | 7          | Methylcyclohexane
CH 3
heptane | 9.26 1
6.23 1 |            |                          |                                     |
| 5 member  | Oxolane                                        | 17.0 1                   | 6          | 2,3-dimethylbutane
Hexane                    | 5,86 1
4.97 1 | 6          | di-n-propyl ether        | 20.0 1                   |
| 4 member  | Oxetane                                        | 10.3 1                   | 5          | 2-methylbutane
Pentane                       | 3.74 1
3.76 1 | 5          | methyl-s-butyl-
ether | 9.67 1                   |
| 3 member  | CHO                                            | 5.93                                | 4          | Butane                                          | 2.381                                  | 4          | 1-metoxypropane          | 9.911                               |
|           | енх
нь с~~о
12ЕВ              | 5.77                                |            |                                                 |                                        |            |                          |                                     |
|           | H₃C → O
tEB                                 | 1.98                                |            |                                                 |                                        |            |                          |                                     |
|           | H 3 C /, CH 3 | 1.81                                |            |                                                 |                                        |            |                          |                                     |
|           | CEB                                            | 1.50                                |            |                                                 |                                        |            |                          |                                     |
|           | oxirane
O                                   | 0,09                                |            |                                                 |                                        |            |                          |                                     |

**(\*) $\times 10^{12}$ units: cm3molecule-1s-1; 1(McGillen et al., 2020)**

R1: Minor Comments: L162: Why are only some reference compounds used for some of the epoxides? Why did you not use all 6 reference compounds?

**Reply from the authors:**

The selection of the reference compounds employed in the kinetic study has been constrained by the experimental technique. Prior to the use of one reference compound, its reference spectrum has been recorded and compared with the reference spectrum of the epoxide compound and methyl nitrite and in a less extent with product spectra resulted from the initiated oxidation of epoxides/reference compound with OH radicals. The comparison should be made to identify the infrared absorption bands which do not overlap with other infrared absorptions features from reaction mixture to allow subtraction procedure without interferences from IR absorptions of other compounds. Another more permissive criterion is to find a reference compound, which has a rate coefficient toward OH radicals not very different than that one of the epoxides. Usually, up to three reference compounds are enough to derive an average reaction rate coefficient.

R1: L169: Keep names the same for compounds (e.g. propene vs propylene) throughout the paper.

**Reply from the authors:**

**The manuscript has been corrected accordingly.**

R1: Figure 1: I think one panel will show the fact that your results are linear adequately. I would recommend putting most of this figure in the SI.

**Reply from the authors:**

Figure 1 represents the proof for each epoxide that our work has been performed in a proper scientific manner with a good quality of the results. The linearity of the plots does not only show that the methods work adequately but provides information of the missing interferences and secondary reactions for each of the investigated epoxides and selected reference compound. We would prefer to keep Figure 1 in the main manuscript and not to move it to the SI.

R1: L339: A range of  $(2.0\pm5.7) \times 10^{-12}$  cm3 molecule-1 s-1 does not make physical sense. The way the error is written includes negative rate constants.

**Reply from the authors:**

This is actually a typo where "-" was replaced by " $\pm$ ". The correct sentence is "the range of (2.0-5.7) ×10-12 cm3 molecule-1 s-1". However, this section has been reworked and new data added.

R1: L344: You mention a theoretical structural analysis but have no basis sets or computational details in the methods for this.

**Reply from the authors:**

The sentence has been removed from the article. We did not perform theoretical calculations in the present work. However, the discussion section has been reorganised and rewritten following the reviewer's suggestions.

R1: Section 4: There are more recent references for the 24-hr average concentration of OH, Cl, and peak Cl atom concentrations during CalNex that probably should be used: Young et al. ACP 10.5194/acp-14-3427-2014 Wang et al. ACP 10.5194/acp-19-3981-2019 Lelieveld et al. ACP 10.5194/acp-16-12477-2016

**Reply from the authors:**

The new 24-hr average concentration of OH, Cl, and peak Cl atom has been assumed and the proper lifetimes have been re-calculated in Table 5

R1: L385: How does the timescale for aerosol partitioning compare to oxidation by OH?

**Reply from the authors:**

No studies have been reported so far for secondary organic aerosol formation from the OH radical initiated oxidation of the epoxides compounds studied in the present work. However, recent studies show that under determinated ph conditions, most epoxides will have time to react on SOAs (particularly those with tertiary carbon atoms) (Minerath et al 2009). We have discussed the atmospheric implications of epoxides in a large context including the most important isoprene derived epoxides. The manuscript has been modified accordingly.

**Changes in the manuscript:**

Line 478: In the laboratory and ambient SOA, epoxides are considered as potential precursors for sulphate esters, polyols, hydroxy nitrates and halides. Epoxide concentrations in the aerosol composition will be limited by the rates of the gas uptake process, which will be largely determined by the gas phase epoxide concentration levels (Minerath and Elrod, 2009; Minerath et al., 2009), the uptake process and particles acidity. Higher acidity of aerosol particles would produce more efficient heterogeneous reactions of epoxides. A recent study suggests that heterogeneous reaction of some epoxides on acidic aerosols are faster and more efficient than gaseous reactions with atmospheric oxidants, and thus these reactions could represent the major removal pathway for epoxides (Lal et al., 2012). Besides, in urban environments the particle acidity can be significantly high, with pH between 0-5. Under such conditions, epoxides can react rapidly, and the reaction products of the heterogeneous reactions could contribute to the growth of the SOA particles and to the possible modification of their physical and chemical properties (Zhang et al., 2007). Minerath et al., (2009) has shown that 1,2-epoxybutane, which possess slower hydrolysis, has an increasing lifetime with increasing pH. It is expected that even at higher pH most of the epoxides will have time to react on SOAs. The monitoring and evaluating of emissions of this kind of compounds in different ambient conditions, as pH and different regimens of NOx, is recommended in rural and urban locations.

R1: L400-401: There should not be any new information in the conclusion - potential future work should not be included.

**Reply from the authors:** The manuscript has been revised accordingly.**

R1: Technical Comments: The L in 480L or 1080L should be consistently formatted through the paper (i.e. capitalization, italicization, space after the numbers, etc.)

**Reply from the authors:** The manuscript has been modified accordingly.

R1: Numbering of equations is inconsistent.

Reply from the authors: The numbering of equations and reactions has been corrected accordingly.

R1: There are subject/verb disagreements, grammar and punctuation errors, and extra spaces and carriage returns that should be checked for.

**Reply from the authors:** The manuscript has been thoroughly checked for grammar and English errors as well as for technical errors.

---

## Author Comment (AC2)

Response to the reviewer's comments for manuscript acp-2021-962

Title: "**Kinetic Study of the Atmospheric Oxidation of a Series of Epoxy Compounds by OH Radicals**"

by Carmen Maria Tovar et al., Atmos. Chem. Phys. Discuss., https://doi.org/10.5194/acp-2021-962-RC1, 2022

Special Issue: Simulation chambers as tools in atmospheric research (AMT/ACP/GMD inter-journal SI)

We are very grateful for the valuable suggestions and comments of both reviewers, which were carefully considered and helped to improve the quality of our manuscript. The original comments from the reviewers are shown in black. Our answers are marked in blue while changes performed in the body of the manuscript are marked in red.

**B. Comments by Referee #2**

R2: Review of: Kinetic Study of the Atmospheric Oxidation of a Series of Epoxy Compounds by OH Radicals This paper summarizes several experiments in a pair of relative rate reactors to determine the rate constants of the OH reactions with a series of epoxides. The kinetics of epoxide reactions with OH have only been studied with a limited number of epoxides to this point. The literature values of those rate constants are compared, and are in good agreement with the values obtained in this study. This paper fills a gap in the literature as the kinetics of these reactions are currently understudied. The authors do a good job in their introduction outlining the importance of this class of compounds and how they are becoming more important through their role in carbon capture technologies. There are a few technical issues that need to be resolved in the specific comments below. My two large issues are as follows. There is a large section of the results/discussion dedicated to hybridization and ring strain. While interesting from a purely physical chemistry point of view I am not sure how much it actually adds to the paper and might be a bit too physical chemistry for the atmospheric chemistry community at large. Were this a J. Phys Chem. submission I think it would be fine, but I'm not sure about ACP. I think it would be more useful if the authors spent a little more time suggesting improvements/modifications to the Structure Activity Relationship (SAR) instead. This paper clearly demonstrates a scenario where SAR is lacking. It would be more informative to say something beyond "we need more data to properly capture the behaviour of the epoxide reactions with OH." Now it is possible that improvements to the five suggested methods are not possible, the authors could explain why this is the case. Overall, the paper fits within the scope of ACP and I recommend publication once the Results/Discussion section has been reworked and the technical issues below are addressed.

**Reply from the authors:**

We very gratefully for the reviewer's comments. The reviewer clearly emphasize that there is a lack of knowledge for the kinetics of epoxide reactions with OH radicals. The new SAR method is beyond the scope of this paper but, as the reviewer mentioned, we intended to demonstrate where SAR is lacking and to provide with our work additional rate coefficients to improve the kinetic data base regarding the gas-phase reactivity of epoxides toward OH radicals.

As we already mentioned in a previous answer, the experimentally obtained rate coefficients from the present study together with existing literature data would not provide enough information to improve current SAR methods and more data are required to properly capture the behaviour of the epoxide reactions with OH radicals. However, more information has been added into the manuscript related to the comparison of the experimental vs SAR estimated rate coefficients. Furthermore, the discussion section has been improved with correlation equations based on the log-log correlation fitting data between rate coefficients for the reactions with OH and chlorine atoms. Besides, the manuscript has been modified by adding a new column in table 2 to emphasize the differences between various SAR methods and experimental rate constants for the reaction of epoxides with OH radicals. It is evident from Table 2 those different methods will work better for some epoxides than others and no SAR approach will cover satisfactorily all the epoxides investigated in present study.

Changes in the manuscript/supplementary text:

Changes in the manuscript:

**Table 3.** Comparison of the rate coefficients for the reaction of OH radicals with the epoxides investigated in this study with the literature data and SAR predictions. $k_{Epoxide}/k_{SAR}$

| Epoxide | $k_{SAR}$ | $k_{Epoxide}/k_{SAR}$ | $k_{Epoxide}$ (cm³ molecule⁻¹ s⁻¹) | Reference |
|---|---|---|---|---|
| Cyclohexene Oxide | [a]$6.37\times10^{-12}$ | 0.93 | $(5.93\pm1.13)\times10^{-12}$ | This work |
|  | [b]$6.14\times10^{-12}$ | 0.97 | $(6.51\pm0.65)\times10^{-12}$ | (El Othmani et al., 2021a) |
|  | [c]$6.31\times10^{-12}$ | 0.94 |  |  |
|  | [d]$5.81\times10^{-12}$ | 1.02 |  |  |
|  | [e]$1.02\times10^{-11}$ | 0.58 |  |  |
| Ethylene Oxide | [a]$2.80\times10^{-13}$ | 0.31 | $(0.88\pm0.25)\times10^{-13}$ | (Calvert et al., 2011) |
|  | [b]$1.89\times10^{-13}$ | 0.47 |  |  |
|  | [c]$3.54\times10^{-13}$ | 0.25 |  |  |
|  | [d]$0.77\times10^{-13}$ | 1.14 |  |  |
|  | [e]$8.85\times10^{-14}$ | 0.99 |  |  |
| Propylene Oxide | [a]$0.59\times10^{-12}$ | 0.83 | $(0.49\pm0.52)\times10^{-12}$ | (Wallington et al., 1988b) |
|  | [b]$0.46\times10^{-12}$ | 1.07 | $(0.30\pm1.00)\times10^{-12}$ | (Middala et al., 2011) |
|  | [c]$0.67\times10^{-12}$ | 0.73 | $(0.47\pm0.24)\times10^{-12}$ | (Virmani et al., 2020) |
|  | [d]$0.28\times10^{-12}$ | 1.75 |  |  |
|  | [e]$5.43\times10^{-12}$ | 0.90 |  |  |
| 1,2-epoxyhexane | [a]$4.64\times10^{-12}$ | 1.24 | $(5.77\pm0.83)\times10^{-12}$ | This work |
|  | [b]$4.48\times10^{-12}$ | 1.29 |  |  |
|  | [c]$4.15\times10^{-12}$ | 1.39 |  |  |
|  | [d]$3.97\times10^{-12}$ | 1.45 |  |  |
|  | [e]$6.34\times10^{-12}$ | 0.91 |  |  |
| 1,2-epoxybutane | [a]$1.81\times10^{-12}$ | 1.09 | $(1.98\pm0.29)\times10^{-12}$ | This work |
|  | [b]$1.65\times10^{-12}$ | 1.20 | $(1.90\pm0.67)\times10^{-12}$ | (Wallington et al., 1988a) |
|  | [c]$1.83\times10^{-12}$ | 1.08 | $(2.20\pm0.02)\times10^{-12}$ | (El Othmani et al., 2021b) |
|  | [d]$1.45\times10^{-12}$ | 1.37 |  |  |
|  | [e]$2.96\times10^{-12}$ | 0.67 |  |  |
| *trans*-2,3-epoxybutane | [a]$0.92\times10^{-12}$ | 1.97 | $(1.81\pm0.33)\times10^{-12}$ | This work |
|  | [b]$0.73\times10^{-12}$ | 2.48 |  |  |
|  | [c]$0.99\times10^{-12}$ | 1.83 |  |  |
|  | [d]$0.49\times10^{-12}$ | 3.69 |  |  |
|  | [e]$9.98\times10^{-13}$ | 1.81 |  |  |
| *cis*-2,3-epoxybutane | [a]$0.92\times10^{-12}$ | 1.63 | $(1.50\pm0.28)\times10^{-12}$ | This work |
|  | [b]$0.73\times10^{-12}$ | 2.05 |  |  |
|  | [c]$0.99\times10^{-12}$ | 1.52 |  |  |
|  | [d]$0.49\times10^{-12}$ | 3.06 |  |  |
|  | [e]$9.98\times10^{-13}$ | 1.50 |  |  |

Calculated using the SAR estimation of (a) (Kwok and Atkinson, 1995), (b) (Calvert et al., 2011), (c) (Calvert et al., 2015), (d) (Middala et al., 2011), (e) (Jenkin et al., 2018).

**R2:** Specific Comments: P4 L122: Were there any losses of reactants on their introduction into the chamber? If so how was this determined?

**Reply from the authors:**

There are no wall losses for neither the epoxides nor the reference compounds. Preliminary tests for potential photolysis have been also performed and no loss by photolysis has been observed over a period of 15 minutes.

**R2:** P5 L147: Perhaps I merely missed this in the description of the chambers, but why did the concentrations have to be 8 times higher in the smaller chamber? Is this a lamp intensity issue with the different chamber materials?

**Reply from the authors:**

The concentrations are approximately 8 times higher in the smaller chamber because there is a similar ratio between the optical path lengths of the White type optical system in the chamber. In order to have similar absorbance with respect to Lambert-Beer law there should be an increase in concentration when the optical path is decreasing. For methyl nitrite the higher concentration would be necessary to provide a high enough steady-state OH radical concentration to react with both the epoxides and reference compounds, respectively.

**R2:** Table 1: There are formatting issues here. The column headers do not really describe what is in the columns. Column 1 is the species list not the initial mixing ratio. This needs to be fixed. What are the uncertainties on the initial conditions? This does not seem to stated anywhere and should be included in this table.

**Reply from the authors:**

We have already answered replied to a similar comment from reviewer 1. The header of table 1 has been modified. We agree that usually the concentrations (mixing ratios in Table 1) should be represented by a value with a range of uncertainties. However, the mixing ratios in table 1 are indicative and are not included in any calculations. A sentence has been introduced into the text to show how these mixing ratios has been obtained. We follow in our determinations the subtraction factor of a reference IR spectrum from all IR spectra recorded during a kinetic experiment. For equation (I) there is a direct dependence between the ratios of concentrations with the ratio of subtraction factors and such later ratios are considered in the relative rate coefficient determinations.

**R2:** P6 L175: Was the linearity only for some compounds or all compounds? This sentence makes it seem like the relationship is not linear for all of the epoxides. This should be reworded, or the lack of linearity needs to be discussed.

**Reply from the authors:**

It is evident from figure 1 that we obtained linear plots with almost "zero" intercept for all compounds investigated in present study. This indicates the missing interferences and secondary reactions and proves that our investigations provide high quality data. The sentence has been re-formulated accordingly: "The linearity has been observed for all the investigated epoxides in both reactors."

**R2:** Table 2: Why were different reference compounds used for the different epoxides? Was it initially merely a spot check on the kinetics of known reactions or something else? This should be mentioned/discussed.

**Reply from the authors:**

The selection of the reference compounds employed in the kinetic study has been constrained by the experimental technique. Prior to the use of one reference compound, its reference spectrum has been recorded and compared with the reference spectrum of the epoxide compound and methyl nitrite and in a lesser extent with product spectra resulting from the initiated oxidation of epoxides/reference compound with OH radicals. The comparison has been made to identify the infrared absorption bands, which do not overlap with other infrared absorptions features from reaction mixture to allow subtraction procedure without interferences from IR absorptions of other compounds. Another more permissive criterion is to find a reference compound, which has a rate coefficient toward OH radicals, which is not very different than that one of the epoxides. Usually, up to three reference compound are enough to derive an average reaction rate coefficient value.

**Changes in the manuscript/supplementary text:**

**Changes in the manuscript:**

Line 170: The selection of the reference compounds employed in the kinetic study has been constrained by the experimental technique since there is required to identify the infrared absorption bands which do not overlap with other infrared absorptions features from reaction mixture.

**R2:** P7 L192 should probably read "such as for cyclic ethers".

**Reply from the authors:**

The sentence has been modified in the revised manuscript

**R2:** P8 L215: The ordering of the SAR methods needs to be changed. The authors jump from (c) to (e) and then come back to (d). This leaves the reader confused and believing they missed something. Figure 1: It is unclear to me which chamber these results came from. It should be indicated in the figure caption and/or panels.

**Reply from the authors:**

Thank you for the comment. The entire Results and Discussion section has been re-written.

Concerning Fig 1: The rate coefficients obtained by using two simulation chambers are in perfect agreement as shown in Fig. 2. Accordingly, the data have been plotted together for all the epoxides regardless the reactor where the experiment has been performed.

**R2:** P10 L243: This is where it would be helpful if the authors made some suggestions or at least discussed a better way to use SAR. Since all of the methods return varying degrees of accuracy for different compounds, it shows that it is not helpful without more data to constrain the SAR prediction. This of course probably is not shocking as with most models if you put either not enough information in (or garbage) you tend to get less than satisfactory results out.

**Reply from the authors:**

We have mentioned as the reviewer suggested that the rate coefficients from the present study together with existing literature data does not provide enough information to improve current SAR methods. All SAR methods report difficulties regarding the agreement of the rate constant with those reported in present study and the literature for the entire series of compounds. Even a factor of 3 has been observed between estimated and observed rate constant values. However, the discussion concerning the factors used in the SAR estimates has been expanded and restructured, in order to have a better view of why structural analysis of epoxides is so important in improving these factors.

**R2:** Table 3: Column 2 needs to have the appropriate letter next to the SAR predicted rate constant. Currently it is up the reader to guess/assume.

**Reply from the authors:**

Table 3 has been modified and the appropriate letter has been placed next to the SAR predicted rate constants.

**R2:** P12 L285: The sentence should not start with brackets. I am guessing this is an Endnote formatting issue that the authors did not catch.

**Reply from the authors:**

We agree with the reviewer's comment. The manuscript has been corrected accordingly.

**R2:** P13 L315: Perhaps it is my own ignorance but I would not have guessed that an epoxide would be less reactive to OH than its analogous alkane. This is a comment more on me than the authors.

**Reply from the authors:**

The discussion derived from Table 4 has been re-written in the revised manuscript and more information has been added.

**R2:** P14 L339: This uncertainty is wrong. How can you potentially have a negative rate constant?

**Reply from the authors:**

This is actually a typo where "-" was replaced by "±". The correct sentence is "the range of (2.0-5.7) ×$10^{-12}$ $cm^3$ molecule$^{-1}$ s$^{-1}$". However, this section has been reworked and new data added.

**R2:** P14 L357: Are the chambers used in these experiments capable of temperature dependant measurements? How stable is the temperature in the reactor over the course of an experiment as I am guessing the lamps do add some heat.

**Reply from the authors:**

The chambers have a permanent monitoring system of the temperature. All experiments have been performed in the range of (298±3) K. The lamps, placed evenly around the chamber, could heat up the air inside the reactor if long time experiments are performed. However, our experiments have been performed over a time period of around 30 min irradiation time, not long enough to heat the chamber over the indicated temperature range.

The 1080 L chamber has an additional temperature control system which could maintain the constant temperature from 278K to 341K and usually is used for temperature dependent kinetic studies. However, the system was not activated during the experiments since the temperature was not out of the (298±3) K range.

**R2:** P15 L372: Should be Cl atoms or radicals?

**Reply from the authors:** We would like to keep "chlorine atoms". This is very often used in the scientific community, although it is well known that the chlorine atom could be a significant reactive free radical in the troposphere (Ravishankara, 2009).

**R2:** P15 L382: What is the aerosol uptake rates and how do they compare to oxidation by OH and Cl? A reference to this should be included.

**Reply from the authors:**

We have discussed the epoxides atmospheric implication in a large context including the most important isoprene derived epoxides. We have already pointed out in the revised version of the manuscript that Minerath et al. (2009) have shown the effect of particles acidity on the 1,2-epoxy butane uptake. The text been modified accordingly in the revised manuscript.

**Changes in the manuscript/supplementary text:**

**Changes in the manuscript:**

Line 478: In the laboratory and ambient SOA, epoxides are considered as potential precursors for sulphate esters, polyols, hydroxy nitrates and halides. Epoxide concentrations in the aerosol composition will be limited by the rates of the gas uptake process, which will be largely determined by the gas phase epoxide concentration levels (Minerath and Elrod, 2009; Minerath et al., 2009), the uptake process and particles acidity. Higher acidity of aerosol particles would produce more efficient heterogeneous reactions of epoxides. A recent study suggests that heterogeneous reaction of some epoxides on acidic aerosols are faster and more efficient than gaseous reactions with atmospheric oxidants, and thus these reactions could represent the major removal pathway for epoxides (Lal et al., 2012). Besides, in urban environments the particle acidity can be significantly high, with pH between 0-5. Under such conditions, epoxides can react rapidly, and the reaction products of the heterogeneous reactions could contribute to the growth of the SOA particles and to the possible modification of their physical and chemical properties (Zhang et al., 2007).

Minerath et al., (2009) has shown that 1,2-epoxybutane, which possess slower hydrolysis, has an increasing lifetime with increasing pH. It is expected that even at higher pH most of the epoxides will have time to react on SOAs. The monitoring and evaluating of emissions of this kind of compounds in different ambient conditions, as pH and different regimens of NO$_x$, is recommended in rural and urban locations.

**R2:** P16 L400: The presentation of future work should either be put in a "Future Work" section or tacked on to the results discussion not the Conclusions section.

**Reply from the authors:** The suggestion of the reviewer has been taken into account and the manuscript has been modified accordingly